# Assessing the co-variability of DNA methylation across peripheral cells and tissues: Implications for the interpretation of findings in epigenetic epidemiology

Eilis Hannon[1], Georgina Mansell[1], Emma Walker[1], Marta F. Nabais[1,2], Joe Burrage[1], Agnieszka Kepa[3], Janis Best-Lane[4,5], Anna Rose[6], Suzanne Heck[7], Terrie E. Moffitt[3,8,9,10], Avshalom Caspi[3,8,9,10], Louise Arseneault[3], Jonathan Mill[1]*

1 University of Exeter Medical School, University of Exeter Medical School, University of Exeter, Exeter, United Kingdom, 2 Institute for Molecular Bioscience, The University of Queensland, Brisbane, Australia, 3 Social, Genetic and Developmental Psychiatry Centre, Institute of Psychiatry, Psychology and Neuroscience, King's College London, London, United Kingdom, 4 Section of Anaesthetics, Pain Medicine and Intensive Care Medicine, Department of Surgery and Cancer, Imperial College London and Imperial College Healthcare NHS Trust, London, United Kingdom, 5 Imperial Clinical Trials Unit, Imperial College London, London, United Kingdom, 6 BRC Flow Cytometry Platform, NIHR GSTT/KCL Comprehensive Biomedical Research Centre, Guy's and St Thomas' NHS Foundation Trust, London, United Kingdom, 7 Biomedical Research Centre at Guy's and St Thomas' Hospitals and King's College London, Guy's and St Thomas' NHS Foundation Trust, London, United Kingdom, 8 Department of Psychology and Neuroscience, Duke University, Durham, United States of America, 9 Center for Genomic and Computational Biology, Duke University, Durham, United States of America, 10 Department of Psychiatry and Behavioral Sciences, Duke University, Durham, NC, United States of America

* J.Mill@exeter.ac.uk

**Data Availability Statement:** The DNAm data are publically available via GEO (accession number GSE166844).

## Abstract

Most epigenome-wide association studies (EWAS) quantify DNA methylation (DNAm) in peripheral tissues such as whole blood to identify positions in the genome where variation is statistically associated with a trait or exposure. As whole blood comprises a mix of cell types, it is unclear whether trait-associated DNAm variation is specific to an individual cellular population. We collected three peripheral tissues (whole blood, buccal epithelial and nasal epithelial cells) from thirty individuals. Whole blood samples were subsequently processed using fluorescence-activated cell sorting (FACS) to purify five constituent cell-types (monocytes, granulocytes, CD4+ T cells, CD8+ T cells, and B cells). DNAm was profiled in all eight sample-types from each individual using the Illumina EPIC array. We identified significant differences in both the level and variability of DNAm between different sample types, and DNAm data-derived estimates of age and smoking were found to differ dramatically across sample types from the same individual. We found that for the majority of loci variation in DNAm in individual blood cell types was only weakly predictive of variance in DNAm measured in whole blood, although the proportion of variance explained was greater than that explained by either buccal or nasal epithelial samples. Covariation across sample types was much higher for DNAm sites influenced by genetic factors. Overall, we observe that DNAm variation in whole blood is additively influenced by a combination of the major blood cell types. For a subset of sites, however, variable DNAm detected in whole blood can be

**Funding:** This work was supported by grants from the UK Medical Research Council [G1002190 (to LA) and K013807 (to JM)], National Institute of Child Health and Human Development [HD077482 to AC], American Asthma Foundation [Senior Researcher Award to JM] and by the Jacobs Foundation (to TEM). Data analysis was undertaken using high-performance computing supported by a MRC Clinical Infrastructure award (M008924) to JM. LA is a Mental Health Leadership Fellow for the UKRI Economic and Social Research (ESRC).The funders had no role in study design, data collection and analysis, decision to publish, or preparation of the manuscript.

**Competing interests:** The authors have declared that no competing interests exist.

attributed to variation in a single blood cell type providing potential mechanistic insight about EWAS findings. Our results suggest that associations between whole blood DNAm and traits or exposures reflect differences in multiple cell types and our data will facilitate the interpretation of findings in epigenetic epidemiology.

## Author summary

As epigenetic variation is cell-type specific, an ongoing challenge in epigenetic epidemiology is how to interpret studies performed using bulk tissue (for example, whole blood) which comprises a mix of different cell types. In this study, we identified major differences in DNA methylation (DNAm) across multiple peripheral tissues and different blood cell types, with each sample type being characterized by a unique signature across multiple genomic loci. We demonstrate how these differences influence commonly used prediction scores derived from DNAm data for age and tobacco smoking, with estimates for the same individual being highly variable across tissues and cell types. Our results enabled us to assess the extent to which variable DNAm in each individual blood cell type relates to variation measured in whole blood. We found that although individual blood cell types predict more of the variation in DNAm in whole blood compared to buccal and nasal epithelial cells, the actual proportion of variance explained is relatively small, except for at sites where DNAm is under genetic control. Our data indicate that for most sites variation in multiple blood cell types additively combines to drive variation in DNAm measured in whole blood. Of note, for a subset of sites, variation in DNAm detected in whole blood can be attributed to a specific blood cell type, potentially facilitating the interpretation of EWAS findings.

## Introduction

There is increasing interest in the role of epigenetic variation in health and disease, with the primary focus of epigenetic epidemiology being on variation in DNA methylation (DNAm) measured in whole blood[1]. Epigenome-wide association studies (EWAS) have been made possible by the availability of high-throughput profiling methods (e.g. the Illumina EPIC microarray[2]) which profile DNAm at specific sites across the genome and can be applied to large numbers of samples. Statistical comparisons are then made by testing for significant differences in DNAm associated with an outcome or exposure. A critical issue for epigenetic epidemiology is that, unlike germline genetic variation, DNAm signatures are tissue- and cell type-specific[3] and the origin of samples used for epigenetic profiling has implications for any conclusions made from these studies. A major caveat of profiling DNAm in DNA isolated from 'bulk' tissues, such as whole blood, is that these represent a heterogeneous mix of different cell types, with the resulting DNAm estimate being an aggregate of the profiles of the constituent cell types weighted by their abundance. As the proportion of individual cell types within a sample can vary across individuals, systematic differences in cellular proportions between cases and controls may manifest as differences in the overall epigenetic profile[3]. Existing studies have addressed this potential confounder by adjusting analyses with covariates capturing the cellular composition of each sample[3]. While this may prevent false positive associations, it does not enable the identification of cell-type-specific variation in DNAm associated with disease or exposure. In addition, subtle changes or differences in rarer cell types

may go undetected as they compete against the background of additional variation from analysing multiple cell types simultaneously. Despite the limitations of analysing DNAm measured in bulk tissue, significant differences in DNAm have been robustly associated with a range of exposures and phenotypes—including tobacco smoking[4–7], body mass index[8,9], autoimmune disorders[10,11] and psychiatric disorders [12,13]–although it is unclear whether trait-associated variation is driven by the variability of DNAm in specific blood cell types.

For practical reasons, most EWAS have been performed using DNA isolated from easily-accessible peripheral tissues (e.g. whole blood or buccal epithelial tissue); although these are often not the primary tissue-/cell type relevant to the phenotype under study, their use facilitates the analysis of the large sample numbers required to overcome the multiple testing penalty inherent in comparing DNAm across ~850,000 sites in the genome[14]. Importantly, intra-individual epigenetic differences (i.e. differences in DNAm occurring at specific sites between tissues and cells within a single person) are larger than inter-individual differences within a specific tissue-type[15–19]. Epidemiological studies using accessible tissues may still be mechanistically informative if inter-individual variation in DNAm is correlated across tissues, although previous studies have shown that inter-individual variation in whole blood is only weakly predictive of inter-individual variation in inaccessible tissues such as the brain for the majority of DNAm sites[15,20]. Furthermore, the choice of peripheral tissue is important, as patterns of covariation are tissue specific and not entirely shared between blood, buccal cells and saliva[21].

In addition to identifying loci where aberrant DNAm is associated with a disease, an increasing application of DNAm data in epigenetic epidemiology is to derive estimates of biological age and certain environmental exposures (most notably smoking status[4,5]). The basis of most of these algorithms is a weighted sum of DNAm levels across multiple sites that is associated with the trait of interest. Given the dramatic differences in both the level of DNAm but also the variance of DNAm between tissues, the accuracy of these predictors maybe highly dependent on the cellular origin of the training DNAm data. To circumvent this issue, some of these algorithms have been developed using data from a wide range of tissues; a major strength of the "Epigenetic Clock", for example, is its ability to estimate age across multiple tissue- and cell types[22]. However, while, tissue agnostic algorithms have been shown to correlate strongly in multiple tissues, the performance of these algorithms in samples collected from different cell and tissue types from the same person, at the same time, has not been extensively assessed.

Our understanding of how variation in DNAm within individual blood cell types contributes to variability in DNAm profiles generated from whole blood is limited by the paucity of data from purified cell types across multiple individuals. In this study, we used the Illumina EPIC microarray to quantify DNAm across the genome in matched DNA samples isolated from buccal epithelial cells, nasal epithelial cells, whole blood and five major blood cell types (monocytes, granulocytes, CD4$^+$ T cells, CD8$^+$ T cells and B cells) from 30 donors. A key aim of our study was to explore how variable DNAm in specific blood cell types reflects methylomic variation in whole blood and the consequences of this for interpreting analyses of DNAm data in epidemiological studies. To this end, first we characterized genome-wide differences in DNA methylation between purified blood cell types and whole blood, assessing how these differences influence commonly used algorithms derived from DNAm data. Second, we characterized patterns of covariation between whole blood and each blood cell type, focusing on identifying sites where variation in DNAm in whole blood reflects variation within a single blood cell type. In order to benchmark our findings, we compared our results with similar analyses performed across tissues, by additionally comparing whole blood with two other peripheral tissues, buccal epithelial cells and nasal epithelial cells. In addition to providing the

research community with a resource to help facilitate the interpretation of EWAS analyses performed in whole blood, our data represent useful reference profiles for whole blood cellular composition deconvolution algorithms and are available from the Gene Expression Omnibus (GEO) repository for this purpose (accession number GSE166844).

## Results

### Distinct genome-wide DNAm profiles drive peripheral cells to cluster by tissue then cell type

Following pre-processing, normalization and stringent quality control (see **Materials and Methods**) our final dataset included measures of DNAm at 784,726 autosomal sites across 217 individual DNA samples isolated from 30 individuals derived from fifteen pairs of monozygotic twins (see **S1 Table** for a summary of demographic measures). In total we generated 76 DNAm profiles from peripheral tissues (28 buccal epithelial, 19 nasal epithelial, and 29 whole blood) and 141 DNAm profiles from purified blood cell types (28 monocytes, 29 granulocytes, 28 CD4+ T cells, 28 CD8+ T cells, and 28 B cells). In order to provide an overview of the similarities and differences in genome-wide DNAm profiles between different sample types, hierarchical clustering analysis was performed using the top 1000 variable DNAm sites (ranked by standard deviation (SD)). As expected, samples clustered almost perfectly by tissue- or cell type, with buccal epithelial cells and nasal epithelial cells showing slightly less-optimal discrimination than observed between other sample types (**Fig 1A**). Across the individual blood cell-types we observed two major clusters, with the first including the B cells and the T cells and the second including the monocytes and granulocytes. There were extreme differences between these groupings with more subtle differences between cell types within each sample group.

To confirm that cell-type differences were the primary drivers of variation in DNAm across samples, principal component (PC) analysis was used to provide representations of the data that captured variation across the full dataset. The first PC, explaining 38.7% of the variance in DNAm across the genome, robustly distinguished buccal epithelial and nasal epithelial samples (both ectodermic) from all the blood-derived samples (mesodermic), reflecting the greater epigenetic differences between tissue lineages than between cell types within a tissue (**S1 Fig**). The second PC, which explained 28.8% of the variance in DNAm, separated T cells and B cells from the other sample types, and the third PC, which explains 8.23% of the variance in DNAm across the genome, separated B cells from the other sample types. There were additional PCs that reflected the more subtle differences in DNAm between closely-related sample types; for example, the fourth PC differentiated buccal epithelial cells from nasal epithelial cells, the fifth PC differentiated granulocytes from monocytes, and the sixth, seventh and eighth PCs differentiated CD4+ T cells from CD8+ T cells. A combination of three PCs (the first, second and sixth) was found to successfully discriminate between the eight individual sample types, demonstrating that each sample type can be characterized by a unique DNAm signature across a subset of autosomal sites included on the Illumina EPIC array (**Fig 1B**).

### Major differences in DNAm at distinct subsets of genomic loci define buccal epithelial, nasal epithelial and blood-derived cells

Having confirmed that there are major differences in genome-wide DNAm profiles between the tissue- and cell- types profiled in this study, we next sought to characterize the distribution of autosomal DNAm levels between sample types. Blood-derived samples had broadly comparable distributions, but showed notable differences compared to both buccal epithelial and

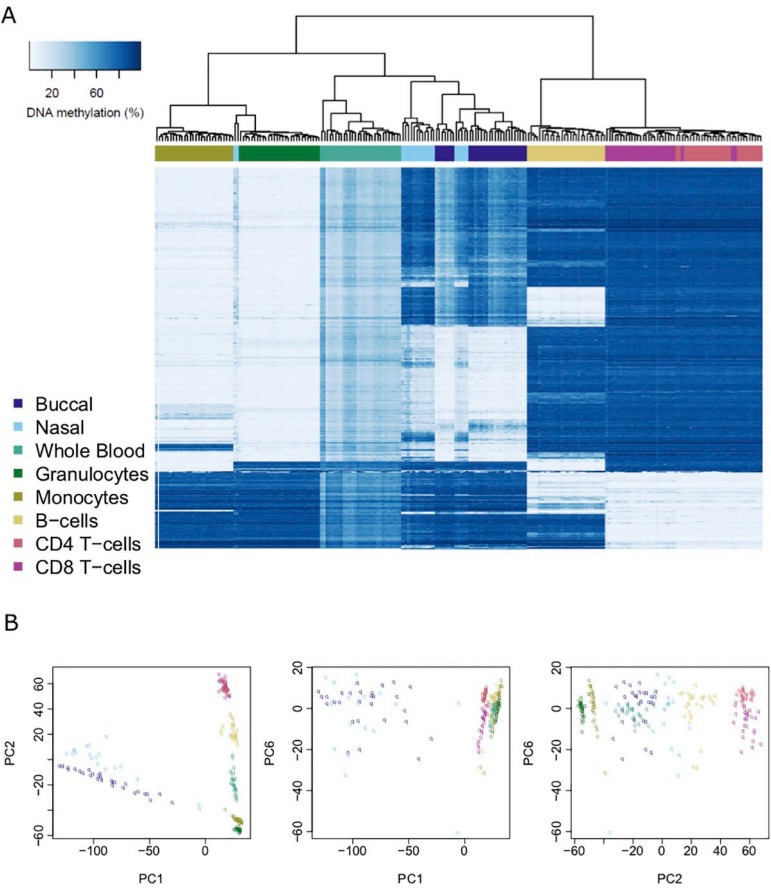

**Fig 1. Major axes of variation in DNA methylation data are driven by sample origin. A**) Heatmap of DNA methylation values across purified cell types and peripheral tissues for the top 1000 most variable sites (ranked by standard deviation). Each row depicts data for an individual DNAm site, and each column depicts data from one individual sample. The order of rows and columns was determined by hierarchical clustering to group together similar profiles of DNA methylation. Low levels of DNA methylation are represented by white boxes and high levels of DNA methylation represented by blue boxes. The colored bars across the top of the columns depict different sample types. **B**) Scatterplots of the three principal components (first, second and sixth) that optimally separate different sample types. Each point represents a sample and the color of the point indicates the specific sample type. The different panels represent different orientations of the same dataset.

nasal epithelial samples (**S2 Fig**). The proportion of sites with low (<20%) and high (>80%) levels of DNAm was lower in buccal epithelial and nasal epithelial cell samples, with a larger proportion of sites characterized by 'intermediate' levels of DNAm (number of sites with mean DNAm between 20 and 80%: buccal epithelial samples = 409,878 (52.2%), nasal epithelial samples = 408,205 (52.0%), blood-derived samples = 358,624 (45.7%)).

To identify specific sites characterized by differential levels of DNAm across sample types we performed an ANOVA of DNAm regressed on sample origin (see **Materials and Methods**). The vast majority of sites (n = 611,070, 77.9%) were characterized by significant (P < 9x10$^{-8}$) differences in DNAm between at least two sample types. As expected, the samples with highest number of sites with significant differences in DNAm relative to our 'reference' tissue (whole blood) were buccal epithelial samples (532,054 DMPs, 87.1%) and nasal epithelial samples (521,660 DMPs, 85.4%) (**S3 Fig**). The fewest number of DNAm differences (294,064 DMPs, 48.1%) was found between whole blood and granulocytes, likely reflecting the fact that granulocytes comprise the most abundant cell type in whole blood. Of the purified blood cell

types, CD8+ T cells had the most sites with differential DNAm compared to whole blood (368,029 DMPs, 60.2%), followed by CD4+ T cells (338,828 DMPs, 55.4%) and B cells (334,359 DMPs, 54.7%). We found that the majority (597,888 DMPs, 97.8%) of these sites were differentially methylated compared to whole blood in more than one tissue or cell type (S4 Fig), although many of these (354,057 DMPs, 59.6%) were characterized by different directions of effect across sample types. Furthermore, each sample type is characterized by only a small proportion of DNAm sites (mean = 1,883 sites, SD = 2,528 sites) that are uniquely different in that sample type alone (S4 Fig). Comparing the DMPs identified for each cell or tissue type— where the mean level of DNAm in that sample type was significantly different relative to whole blood—we found that there was a large overlap (S5 Fig); in other words where DMPs were significantly different in one cell or tissue type, they were also generally different in multiple sample types. As expected, the samples showing the highest overlap in DMPs compared to whole blood were nasal epithelial cells and buccal epithelial cells (468,349 DMPs, 89.8%; S6 Fig), indicating that whole blood is generally different to both of these tissues at a very similar set of DNAm sites. There was also a high overlap between DMPs identified in CD4+ T cells and CD8+ T cells (298,992 DMPs, 81.2%) and also between granulocytes and monocytes (219,037 DMPs, 69.1%), reflecting the consistent developmental lineages of these cell-types. Overall, our results suggest that cell-specific levels of DNAm occur in a hierarchical manner, where DNAm across a larger proportion of the genome is shared between cells from the same linage (S6 Fig). Furthermore, because only a minority of sites can be considered as indicators of specific sample origin across the cell types or tissues tested, our results indicate that more complex combinations of multiple DNAm sites are required to accurately distinguish actual cell types.

## Measures of age and environmental exposure derived from DNAm data differ dramatically across sample types from the same individual

One common application of DNAm data is to calculate predictors for age or certain environmental exposures, although it is unclear whether the accuracy of these predictors differs across sample types. As all our donors were the same age at sample collection (19 years), we can compare the variation in prediction across both individuals and sample types. Our data suggest that 'DNAm age' predicted using the Horvath DNAmAge Epigenetic Clock[22] differs across sample types taken from the same individual at the same time (Fig 2). For example, DNAm data generated in whole blood samples yields the oldest and most inaccurate age predictions (mean difference to actual age = 11.5 years). Performing statistical comparisons between sample types using whole blood as the reference tissue highlighted significantly lower estimates of age when using DNAm data generated using buccal epithelial cells (mean difference compared to whole blood = -11.0 years, $P = 1.65 \times 10^{-15}$) and nasal epithelial cells (mean difference compared to whole blood = -14.3 years, $P = 4.96 \times 10^{-22}$) (S2 Table). We also found significantly lower DNAm age estimates in samples derived from the majority of purified blood cell types relative to whole blood (B cells (mean difference compared to whole blood = -5.78 years, $P = 5.31 \times 10^{-6}$), CD4+ T cells (mean difference compared to whole blood = - 4.22 years, $P = 7.11 \times 10^{-4}$), CD8+ T cells (mean difference compared to whole blood = -8.43 years, $P = 1.80 \times 10^{-10}$), monocytes (mean difference compared to whole blood = -4.79 years, $P = 1.34 \times 10^{-4}$). In contrast, age estimates derived from granulocyte samples were not significantly different to whole blood, likely reflecting the fact that they are the most abundant blood cell type. Of note, the variation in age estimates derived from DNAm data across sample types derived from the same individual (mean range = 16.4 years (SD = 4.29 years)) is notably greater than the range of ages between individuals within a specific tissue or cell type (mean range = 12.1 years, SD range = 4.24 years) and the reported error of the Horvath multi-tissue DNAm clock

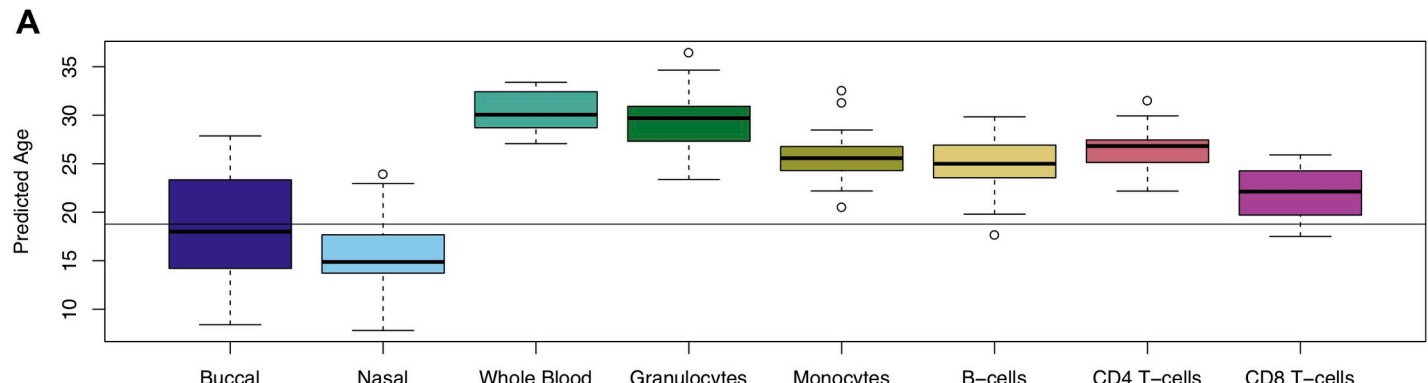

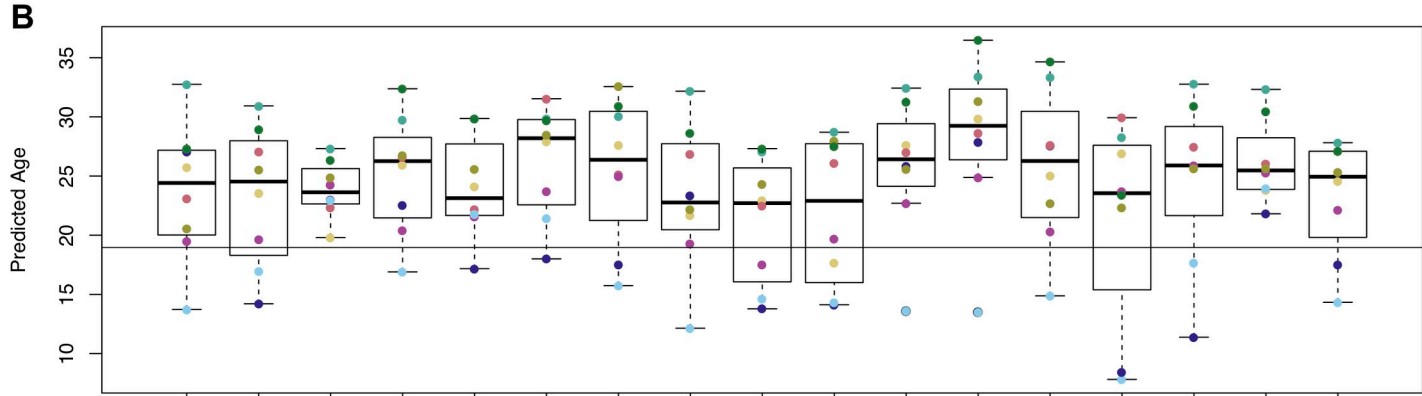

**Fig 2. Age estimates derived from DNA methylation data vary across different sample types from the same individual. A**) Boxplots of predicted age using the Horvath epigenetic clock (Horvath 2013), with samples grouped by sample type. Predictions for each sample type had a mean range of 12.1 years (SD 4.24). **B**) Boxplots of predicted age grouped by individual, where the color indicates the value derived from a specific sample type from that individual. Predictions for each individual had a mean range of 16.4 years (SD = 4.29). The line shows the actual age of the samples (19 years old).

(median absolute deviation = 3.6 years)[22]. We used a similar approach to explore the consistency of a tobacco smoke exposure predictor across different tissues and cell types (**Fig 3**). Our comparison showed that compared to whole blood, samples derived from buccal epithelial cells (mean difference compared to whole blood = 14.0, P = $1.50\times10^{-21}$) and nasal epithelial cells (mean difference compared to whole blood = 12.9, P = $7.35\times10^{-20}$) had higher (and more variable) derived smoking scores (**S3 Table**). Because our samples were collected from a cohort of adolescent donors and the frequency of actual reported smoking was relatively low (20% current smokers; mean = 0.520 pack years; SD = 0.724 pack years) we expected the derived smoking scores to be approximately zero, as seen in the different purified blood cell types (**Fig 3**). It is plausible that the higher derived smoking scores observed in buccal epithelial cells and nasal epithelial cells reflect both passive exposure to tobacco smoke[23] and exposure to other environmental toxins (e.g. air pollution) that influence DNAm at similar sites to smoking[24].

## Differences in the variability of DNAm between buccal epithelial, nasal epithelial and blood-derived samples at multiple loci across the genome

It is well documented that variation in cell proportions between comparison groups (e.g. patients and controls) may introduce apparent differences in DNAm in whole blood samples at specific sites leading to the reporting of false positives in EWAS analyses of health and disease[3]. Previous comparisons between cell types and tissues have primarily focused on

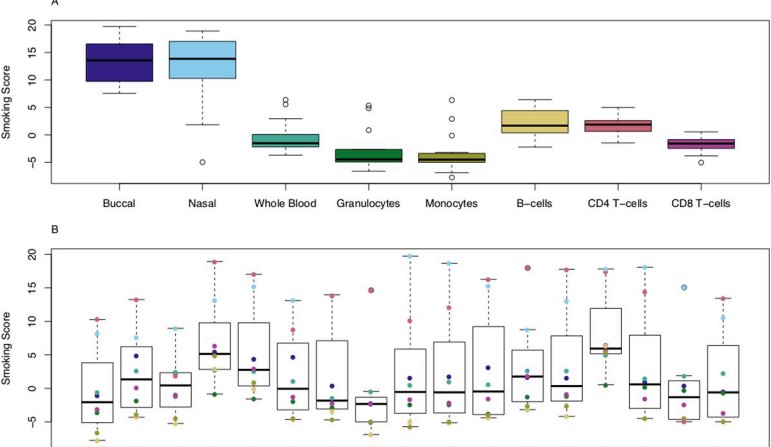

**Fig 3. Smoking scores derived from DNA methylation data vary across different sample types from the same individual. A)** Boxplots of estimated exposure to tobacco smoking grouped by cell or tissue type. Smoking score predictions for each sample-type had a mean range of 12.1 (SD = 4.24). **B)** Boxplots of estimated exposure to tobacco smoking grouped by individual, where the color indicates the value derived from a specific sample type. Smoking score predictions for each individual had a mean range of 11.5 (SD = 6.21).

differences in the actual level of DNAm, with limited information about whether inter-individual variation is correlated across constituent cell types. To investigate this further, we calculated the standard deviation in DNAm level at each site for each sample-type across all individuals, finding that buccal epithelial cell samples (mean SD = 4.44%, SD = 2.85%) and nasal epithelial cell samples (mean SD = 5.41%, SD = 3.69%) are generally more variable compared to any of the blood-derived samples (**S7 Fig**). Although it might be expected that DNAm profiles generated in a bulk tissue, which reflect multiple cell types, would be more variable than those generated in an isolated cell type, we did not always find this to be the case. We observed that DNAm in whole blood (mean SD = 2.57%, SD = 1.67%) is, on average, less variable than DNAm measured in B cells (mean SD = 3.39%, SD = 2.06%), monocytes (mean SD = 2.94%, SD = 1.78%) and CD8$^+$ T cells (mean SD = 2.97%, SD = 1.90%), suggesting that variability present in minor blood cell types makes a relatively small contribution to variability in whole blood. We observed moderate to strong positive correlations (r = 0.39–0.88) between the magnitude of variation in DNAm for all pairs of sample-types tested (**Figs 4** and **S8**). The strongest correlation in DNAm variation was between CD4$^+$ T cells and CD8$^+$ T cells (r = 0.88), although variance in whole blood was strongly correlated with variance in all five cell types (r = 0.73–0.86) and only moderately correlated with variance in buccal (r = 0.52) or nasal (r = 0.49) epithelial samples. Although these correlations indicate that variable DNAm sites identified in one sample type are often variable in other sample types, it does not mean that the actual magnitude of variation is comparable. Given the differences in the distribution of inter-individual variance, we used Levene's test to identify the specific sites where inter-individual variation is significantly different between cell/tissue types. We identified 194,247 'differentially variable positions' (DVPs; P < 9.42x10$^{-8}$) characterized by a significantly different variance in DNAm present in at least one sample type. Comparing the distribution of standard deviations across these DVPs, we observed that the majority of these differences were driven by higher levels of variability in buccal epithelial or nasal epithelial cell samples (**S9 Fig**) compared to the blood-derived samples. Furthermore, looking at the correlation of variance between tissues and cell types at these DVPs, we found that sites characterized by higher variation in buccal epithelial cells generally showed higher variation in nasal epithelial cells, and

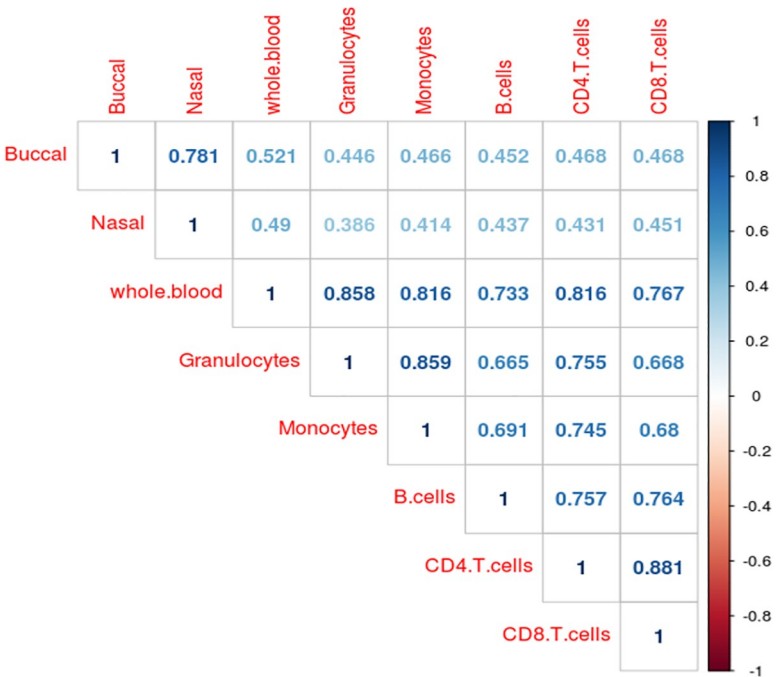

**Fig 4. Inter-individual variation in DNA methylation is correlated between different tissues and cell types.**

vice versa (r = 0.649; **S10** and **S11 Figs**). Interestingly, however, this correlation was smaller than that observed between certain purified blood cell-types including monocytes and granulocytes (r = 0.732) and CD4$^+$ T cells and CD8$^+$ T cells (r = 0.828).

## Inter-individual variation in DNAm is mirrored across tissues and cell types at a relatively small number of sites

Having identified differences in both the level of DNAm and the magnitude of variation of DNAm across sample types, we sought to characterize how DNAm profiles in the five purified blood cell types combine to influence the DNAm profile measured in whole blood. To quantify how inter-individual variation in DNAm in whole blood reflects inter-individual variation in the purified blood cell types, we explored the covariation of DNAm levels between whole blood and each constituent cell type across all individuals. The distribution of correlation coefficients across the genome is skewed to the right of zero, indicating an excess of positive correlations (**S12 Fig**). Although the overall positive skew was expected given that variation in whole blood is driven by the variation in these individual cell types, it is noteworthy that for the majority of DNAm sites inter-individual variation in DNAm in whole blood explained only a small proportion of the variation seen in any single blood cell type (**Fig 5**). For example, variation in DNAm in granulocytes—the most abundant individual cell type in whole blood—explains a mean of 10.97% (SD = 17.8%) of the variance in DNAm in whole blood, with only 11,061 (1.41%) sites where variation in granulocytes alone explained more than 80% of the variation in whole blood. The summary statistics for the four other blood cell types were, as expected, even lower (mean variance explained by monocytes 9.30% (SD = 15.9%); B cells 8.03% (SD = 13.8%); CD4$^+$ T cells 9.95% (SD = 15.9%); CD8$^+$ T cells 9.53% (SD = 14.9%)). Of note, all of the blood-derived cell types explained on average a higher percentage of variation in DNAm in whole blood DNAm than the two additional peripheral tissues we assessed in this study (**Fig 5**). Using inter-individual variation in buccal epithelial cells to predict inter-

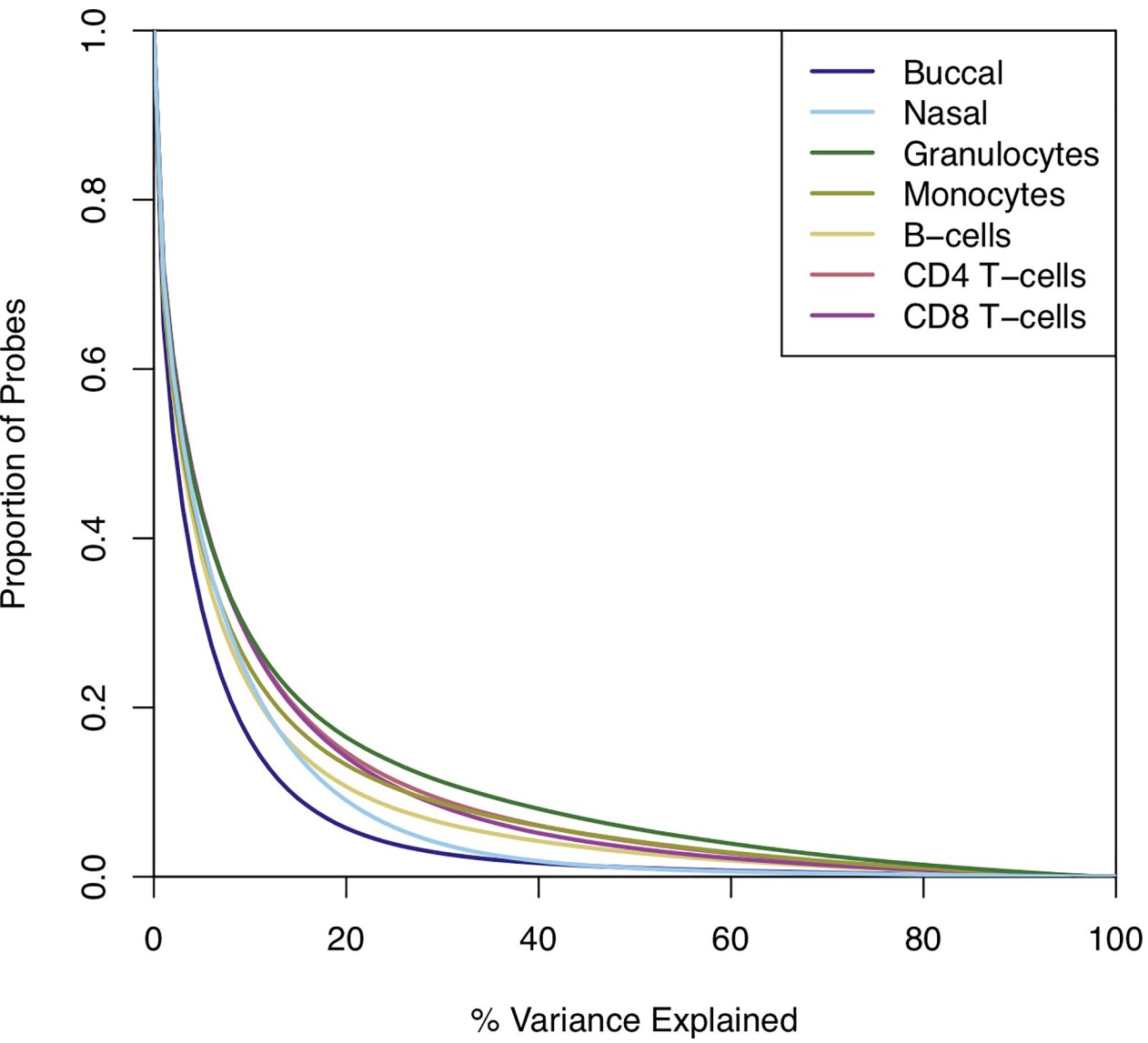

**Fig 5. Variation in DNA methylation in whole blood as a predictor of variation in the five isolated blood cell types and other peripheral tissues (buccal and nasal epithelial cells).** Shown is the proportion of sites (y-axis) for which variation in whole blood DNA methylation explains different levels of variance in additional sample types isolated from the same individuals.

individual variation in whole blood, for example, we found that the mean variance in DNAm explained across all DNAm sites was only 5.64% (SD = 9.80%). Similarly, using inter-individual variation in nasal epithelial cells to predict inter-individual variation in whole blood, we found that the mean variance in DNAm explained across all DNAm sites was 7.12% (SD = 10.5%) (**Fig 5**).

### Covariation in DNAm between sample types is higher at sites with intermediate and variable levels of DNAm, and at sites where DNAm is strongly influenced by genetic effects

Although the number of probes where variation in whole blood DNAm indexes variation in a specific blood cell type was relatively low, we found that covariation in DNAm was higher in

the subset of sites classed as having 'intermediate' levels of DNAm (i.e. mean DNAm between 40 and 50%) or being highly 'variable' (i.e. range of DNAm levels in the middle 80% of sample being > 5%) (S13 Fig). We previously characterized the proportion of variance in whole blood DNAm explained by additive genetic and environmental factors in a large cohort of monozygotic and dizygotic twin pairs, finding larger genetic influences at sites with variable and intermediate levels of DNAm[25]. We used this dataset to explore covariation in DNAm for the subset of sites where whole blood DNAm is under very strong genetic control (additive genetic effects on DNAm > 0.80, n = 6,617 sites), finding dramatically higher covariation in DNAm between whole blood and the individual blood cell-types compared to the levels seen across all sites in our dataset (S14 Fig). Across these sites, variation in DNAm in granulocytes explains a mean of 61.37% (SD = 34.67%) of the variance in DNAm in whole blood, with relatively higher levels also seen for the other individual blood cell types (mean variance explained by monocytes 57.09% (SD = 34.13%); B cells 50.75% (SD = 33.74%); CD4$^+$ T cells 55.0% (SD = 34.87%); CD8$^+$ T cells 51.91% (SD = 34.15%) (S15 Fig). Furthermore, we also found elevated levels of covariation in DNAm between whole blood and individual blood cell types for sites directly associated with a DNAm quantitative trait locus (mQTL) in whole blood using a dataset recently generated by our group[26] (n = 147,683 sites) (S16 and S17 Figs). In contrast, we found that covariation was attenuated at sites where DNAm was strongly influenced by environmental factors (non-shared environmental effects on DNAm > 0.80, n = 143,501 sites) (S18 Fig). Across these sites variation in DNAm in granulocytes explains a mean of only 4.31% (SD = 6.32%) of the variance in DNAm in whole blood, with equally low levels also seen for the other individual blood cell types (mean variance explained by monocytes 4.14% (SD = 5.99%); B cells 4.05% (SD = 5.72%); CD4$^+$ T cells 4.41% (SD = 6.28%); CD8$^+$ T cells 4.46% (SD = 6.23%) (S19 Fig). These results suggest that genetic effects act to increase correlation between cells/tissues while environmental factors act to decrease correlation, presumably because the former is a pan-tissue phenomenon and environmental factors commonly affect only a subset of cells and tissues.

## Variation in DNAm in whole blood is driven by cumulative variance in multiple blood cell types

We next tested the hypothesis that variable DNAm between individuals measured in whole blood reflects additive effects across multiple blood cell types and does not reflect individual cell types exclusively driving variation in DNAm at a different subset of sites. This hypothesis was supported by the observation that covariation between DNAm measured in whole blood and the specific blood cell types was highly correlated across the different cell-types profiled in this study (range in r = 0.539–0.655; S20 Fig). To quantify how much DNAm variation in whole blood the five cell types explain in combination, we fitted a mixed effects linear model for each DNAm site predicting DNAm in whole blood from the DNAm levels measured in the five purified blood cell types simultaneously (S21 Fig). With all five cell types included, the total amount of DNAm variance in whole blood explained is almost double that observed for any individual cell type individually (mean variance explained = 25.7% (SD = 18.5%)). Furthermore, the number of sites where the amount of DNAm variance in whole blood explained is greater than 80% increased to 17,517 (2.23%) when all constituent cell types were considered. Again, the proportion of variance explained was highest in sites with either "variable" (mean variance explained = 30.7%; SD = 21.2%) or "intermediate" levels of DNAm in whole blood (mean variance explained in DNAm sites characterized by levels of DNAm between 40 and 50% = 42.6%; SD = 23.9%) (S22 Fig).

## At a subset of sites across the genome, variation in whole blood DNAm is driven by variation in a single blood cell type

Although variation in DNAm in whole blood reflects variation across multiple cell types for most sites in the genome, at a number of DNAm sites a large proportion of the variation in DNAm ($> 20\%$) is driven by variation in a single cell type (N = 109,405; 13.9%; **S23 Fig**). These sites are of particular interest for epigenetic association studies performed in whole blood because trait-associated variation at these sites might potentially be attributed to changes in a specific cell population, facilitating the interpretation of findings. We therefore sought to quantify the extent to which variation in DNAm measured in whole blood reflects underlying variation in specific blood cell types, generating what we refer to as "characteristic scores" for each blood cell type across all DNAm sites (see **Materials and Methods**). From these characteristic scores, we identify 30,514 sites (3.89%) where variation in DNAm in a single cell type was the dominant driver of variation in DNAm measured in whole blood. Of note, variation at these sites primarily reflected variation in B cells (N = 20,888; 2.66%), although all five blood cell types included in this study were exclusively responsible for variation at a subset of DNAm sites (**S4 Table**). These characteristic scores can be used to annotate whole blood EWAS results to determine which cell types are potentially affected by the significant differences reported. To demonstrate the potential utility of these scores, we annotated differentially methylated sites robustly associated with body mass index (BMI) and smoking from previously published studies[9,27]. Of the differentially methylated sites associated with BMI by Wahl et al, 177 were available in our final dataset, with 27 (15.3%) of these demonstrating characteristic variation of a single cell type (**S5 Table**). Twelve whole blood DMPs associated with BMI reflect variation in B cells, 9 reflect variation in CD8[+] T cells, 4 reflect variation in granulocytes and 2 reflect variation in CD4[+] T cells. Of the differentially methylated sites in whole blood associated with tobacco smoking by Joehanes et al, 16,746 were present in our final dataset with 1,881 (11.2%) of these reflecting variation in a specific blood cell type (**S6 Table**). The vast majority of these (N = 1,286; 68.4%) reflected variation in B cells, with almost a quarter reflecting variation in CD8[+] T cells (N = 437; 23.2%). For both phenotypes, variation in multiple blood cell types appeared to underlie the reported EWAS associations identified in whole blood (**Fig 6**), indicating that analyses in purified populations of cells may need to interrogate multiple cell types to fully characterize the epigenetic consequences of common exposures such as tobacco smoking and BMI. To expand these analyses to DMPs associated with a broader range of traits, we downloaded results from the latest version of online EWAS catalog (http://ewascatalog.org/, accessed on 22/09/2020) and identified 2,920 significant associations ($P < 10^{-7}$) from studies performed in whole blood that are characteristic of a single blood cell type based on our characteristic scores (**S7 Table**). For traits with at least five characteristic sites we calculated a fold change statistic to identify enrichments for particular blood cell types (**S24 Fig**). This revealed some interesting patterns: sex, for example, had 431 associated sites that were characteristic of a single blood cell type with the distribution across blood cell types in line with the expected ratios. In contrast, DMPs associated with C-reactive protein (CRP) levels were enriched for sites characteristic of granulocytes, and DMPs associated with chronic kidney disease were enriched for sites characteristic of monocytes.

## Discussion

In this study we characterized DNAm in multiple tissues and cell-types (whole blood, buccal epithelial cells, nasal epithelial cells, granulocytes, monocytes, B cells, CD4[+] T cells and CD8[+] T cells) derived from thirty individuals. We identified major differences in DNAm between blood cell types and peripheral tissues, with each sample type being characterized by a unique

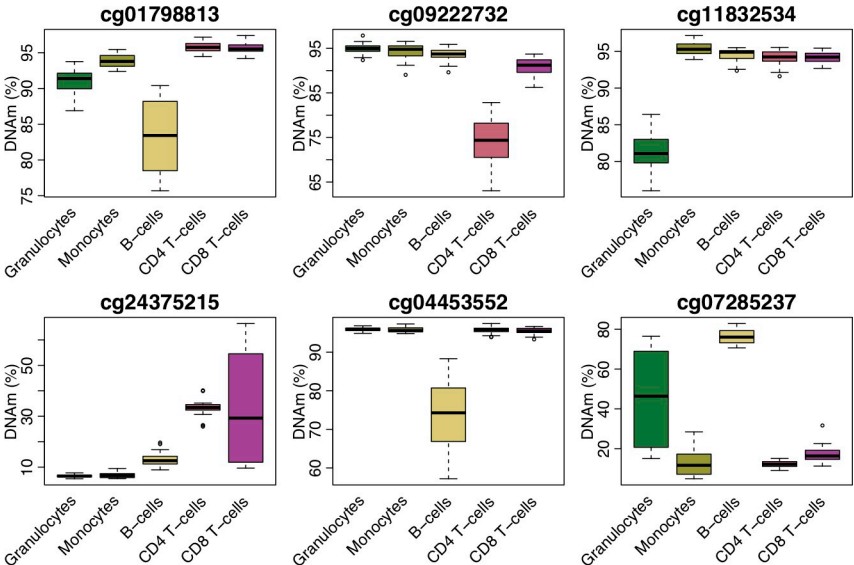

**Fig 6. Trait associated variation in whole blood DNA methylation can often be attributed to a specific cell type.** Boxplots of DNA methylation sites associated with body mass index (BMI) (top row) and tobacco smoking (bottom row) where variation identified in whole blood is attributable to variable DNA methylation in a single blood cell type.

DNAm signature across multiple loci. The number of DMPs identified between sample types (n = 611,070; 77.9% of those tested) is in line with other studies comparing DNAm differences between tissues[28,29] and highlights the importance of accounting for cellular heterogeneity in epigenetic epidemiology. Our study confirms previous results demonstrating that tissue and cell-type is the primary source of variation in DNAm [15,30,31] and that DNAm profiles derived from blood-based samples are strikingly different to those from other tissues[32,33]. Our data show that the identity of different sample types is defined by the presence or absence of DNAm at specific loci, with signatures across different subsets of sites required to distinguish different sample types.

We also show that there are important differences in measures of age and smoking exposure derived from DNAm data across sample types from the same donor. Although all samples were collected from individuals at the same age (age 19), the mean range of predicted ages for an individual across sample types was 11.5 years, highlighting a potentially large degree of error for individual-level predictions depending on the sample type used for epigenetic profiling. Although younger samples were included in the training data used to develop the prediction model, the Horvath clock was largely developed using adult samples, and therefore the reduced accuracy in our samples may be due to reduced representation of similar aged individuals in the development of the epigenetic clock algorithm. Most assessments of its performance have been in cohorts where the sample age is variable and the focus has been on estimating the strength of the prediction against chronological age, where the magnitude of the correlation is in part driven by the variation in chronological ages in the test sample. Although previous studies have alluded to the importance of tissue specificity in the accuracy of pan-tissue epigenetic clocks[34,35], we provide direct empirical evidence to support this notion. Only a limited number of studies have previously explored its performance in cohorts where donors are all the same age, or compared its performance between different sample-types from the same individual. In the original Epigenetic Clock manuscript[22], a number of individuals had epigenetic age estimated from multiple tissues and although the coefficient of variation reported was classed as "low", the range of predicted ages for the same individual was

similar to that observed in this study. Our results do not dispute the utility of the Epigenetic Clock at a study level to distinguish older individuals from younger individuals and approximately rank samples by age but suggest that individual predictions should be interpreted with caution and the absolute age prediction should be considered in the context of the tissue or cell type in which DNAm data was generated. Our data support the importance of developing tissue-specific epigenetic clocks, with several studies highlighting increased accuracy for clocks when used specifically in the tissues they are trained on [36–40].

In addition to comparing DNAm at specific sites between peripheral tissues and individual blood cell types, we also tested for differences in the magnitude of inter-individual variation across sample types. Although previous work has focused on identifying sites where DNAm levels can be used to distinguish tissues or cell types, the primary interest for epigenetic epidemiology are the sites at which DNAm varies within a population. We identified 194,247 sites with significantly different variances in DNAm between peripheral tissues and the major blood cell types isolated in this study, highlighting that sites variable in one tissue or cell type are not necessarily also variable in another. Future studies should look to identify the sources of this cell-type specific variation. Furthermore, we found that variation in one peripheral tissue (e.g. buccal epithelial or nasal epithelial cells) was only lowly predictive of variation in whole blood, comparable to results previously for the covariation of DNAm between whole blood and four brain regions from the same individual[15]. If we assume that the variation across individuals is a consequence (or correlate) of a given factor (e.g. environment, disease or genotype), then our results suggest that these effects may be specific to an individual cell/tissue-type or at the very least have differing magnitudes of effect across cell-types and tissues. This has implications for study design as it reinforces the notion that DNA methylation profiles in one peripheral tissue cannot necessarily be used as a proxy for profile of another peripheral tissue.

As DNAm profiles measured in whole blood essentially reflect the aggregation of profiles from individual blood cell types, weighted by the cellular abundance, we hypothesised that variation in DNAm measured in major blood cell types would explain much of the variation measured in whole blood. While individual blood cell types did predict more of the variation in whole blood compared to buccal epithelial and nasal epithelial cells, the percentage of variance explained was still small. Instead our data indicate that at the majority of sites, variation in multiple blood cell types additively combines to drive variation in DNAm in whole blood. For the majority of sites, therefore, it is not possible to determine the individual cell type that is responsible for trait-associated variation identified in whole blood, highlighting the utility of profiling DNAm in purified cell populations to identify the primary cell type involved. Of note, however, we did identify a subset of sites where we could attribute the variation in DNAm measured in whole blood to variation in a single blood cell type. These results have utility for the interpretation of results from EWAS analyses performed in whole blood, enabling the identification of specific cell-types driving associations and providing hypotheses for future functional work. To demonstrate this, we took significant results from recent EWAS analyses of BMI and tobacco smoking and identified sites where variation in DNAm in whole blood likely reflects variation within a specific cell type.

Our results should be interpreted in the context of the following limitations. First, although this represents the most extensive comparison of purified blood cell types to date, our samples were derived from only thirty individuals. Second, our samples are all the same age (19 years old) and of predominantly European decent; although this enabled us to minimize certain additional drivers of variability in DNAm our results may be not applicable to cohorts with different demographics. To this end we have focused on characterising the overall pattern of results, rather than discussing specific DNAm sites. Third, we only considered five major

blood cell types purified from whole blood, and it is likely that many more blood cell (sub) types exist with varying abundance between individuals and different DNA methylation levels; future studies should include these additional rarer blood cell types that may be driving variation in DNAm in whole blood. Fourth, although the Illumina EPIC array quantifies DNA methylation at sites annotated to the majority of genes, the actual proportion of sites across the genome interrogated by this technology is relatively low (~3%), with a predominant focus on CpG-rich promoter regions. It will be important for future studies to explore tissue and cell type differences in DNAm across regions not well-covered by the EPIC array. Fifth, our analyses were performed using beta-values and therefore our results may only be relevant to EWAS analysed using DNAm quantified as a proportion. M-values have been proposed as an alternative method to overcome issues relating to heteroscedasticity [41], although we chose to focus on beta-values as they are more widely-used in epigenetic epidemiology. Of note, we have previously investigated the potential effect of heteroscedasticity on linear regression models fitted using beta-values, finding that this bias does not lead to an excess of false positives [14]. Importantly, the use of M-values is likely to exaggerate technical variation, which is most prevalent at the extreme ends of the distribution of DNAm levels, consequently introducing false positives in our comparisons of different sample types. Finally, differences in variance observed between sample-types with different mean levels of DNAm do not necessarily reflect biological differences as highlighted in recent QTL analyses [42,43], and this is important to acknowledge when interpreting our analysis of DVPs.

To conclude, we have characterised differences in DNAm between three peripheral tissues (whole blood, buccal epithelial cells and nasal epithelial cells) and five major blood cell types (granulocytes, monocytes, B cells, CD4+ T cells, CD8+ T cells) isolated from the same individuals. We identify numerous differences in the variability of DNAm between tissue and cell types, indicating that differences in DNAm (induced by environmental exposures, for example) might have variable effects across cell types and tissues. Finally, we identify sites for which variable DNAm detected in whole blood can be attributed to variation in a single blood cell type. Taken together, our results indicate that for the majority of sites, it is not possible to determine from an analysis performed in whole blood the specific cell type(s) that any effect is attributable to; therefore, profiling isolating populations of cells is likely to improve our understanding of the mechanisms underlying epigenetic dysregulation. These results provide an informative framework for interpreting associations between differential DNA methylation and complex traits identified in whole blood and reiterate the need for generating cell-specific DNAm profiles in future studies.

## Materials and methods

### Ethics statement

This study was approved by the UK National Research Ethics Service London Committee (15/LO/0155) and informed written consent was given by all participants.

### Sample description

Participants were recruited from the Environmental Risk (E-Risk) Longitudinal Twin Study, which tracks the development of a birth cohort of 1116 British twin pairs ($n$ = 2232 individuals) drawn from a larger birth register of twins born in England and Wales in 1994–1995 [44]. Full details about the larger E-Risk sample are reported elsewhere [45]. For this study a sub-sample of 15 twin pairs ($n$ = 30 individuals) were recalled at age 19 for assessment where each participant simultaneously provided a whole blood sample, a buccal swab sample and a nasal

swab sample. This subsample consisted of 14 monozygotic twin pairs and 1 same sex dizygotic twin pair, with 9 female twin pairs and 6 male twin pairs (S1 Table).

## Isolation of purified cell types from whole blood

10 ml of whole blood collected in EDTA vacutainer tubes was used for the purification of specific blood cells from each individual. We used fluorescence-activated cell sorting (FACS) to successfully obtain purified populations of monocytes, granulocytes, CD4+ T cells, CD8+ T cells, and B cells from whole blood from 29 of the 30 individuals. Briefly, 2 x 5 ml of blood were transferred to two 50 ml conical tubes and red blood cells were lysed by adding 45 ml of 1x BD PharmLyse solution (Becton Dickinson, #555899) and incubating for 15 minutes at room temperature. Cells were spun down at 300xg for 5 minutes, washed once in 0.5ml PBS/ 2% FCS/1mM EDTA and finally the pellet was resuspended in 0.5ml PBS/2% FCS/1mM EDTA. Fc receptors were blocked by incubating the cells with 50 μl Human TruStain FcX (Biolegend, #422302) for 10 minutes at room temperature prior to antibody staining and sorting. 50 μl of unstained cells were resuspended in 1 ml PBS/2% FCS/1mM EDTA for sorting of granulocytes based on FSC/SSC only. DAPI (Sigma) was added to stain dead cells just before sorting at a final concentration of 1μg/ml. The remaining 450μl of cell suspension was stained with 15 μl of each of the following antibodies: CD3 APC (Biolegend, 300411), CD4+ FITC (BD, 555346), CD8+ BV711 (BD, 563677), CD19 PerCPCy5.5 (BD, 561295), CD14 APC-Cy7 (BD, 557831). A summary of the number of mean number of individual blood cell-types collected from whole blood is given in S8 Table. Cells were incubated with the antibodies for 30 minutes at room temperature in the dark. Cells were then washed with 5 ml PBS/2% FCS/ 1mM EDTA, spun at 300xg for 5 minutes and resuspended in 3ml PBS/2% FCS/1mM EDTA. DAPI (Sigma) was added to stain dead cells just before sorting at a final concentration of 1 μg/ ml. Sorting was performed on a 5 laser BD FACSAriaII (Beckton Dickinson) using a 100 μm nozzle to sort granulocytes and a 70 μm nozzle for all other cell types. Cell fractions were collected in 5ml round bottom FACS tubes (Falcon), spun down at 300xg for 5 minutes and the pellet was frozen at -80C for later DNA isolation.

## Genome-wide quantification of DNA methylation

Genomic DNA was extracted from the eight sample types (whole blood, buccal, nasal, monocytes, granulocytes, CD4+ T cells, CD8+ T cells, and B cells) using the Qiagen AllPrep RNA/ DNA kit (Qiagen, CA, USA). 500ng of DNA from each sample was treated with sodium bisulfite, using the EZ-96 DNAm-Gold kit (Zymo Research, CA, USA). DNAm was quantified using the Illumina Infinium HumanMethylationEPIC BeadChip (Illumina Inc, CA, USA) run on an Illumina iScan System (Illumina, CA, USA) using the manufacturers' standard protocol. Samples from the same individual were processed together across all experimental stages to negate any methodological batch effects.

## Illumina array pre-processing and quality control

All quality control and statistical analyses were performed using the statistical language R (version 3.4.3)[46]. Raw data for 234 samples (comprising 30 buccal, 30 nasal, 29 whole blood, 29 monocytes, 29 granulocytes, 29 CD4+ T cells, 29 CD8+ T cells, and 29 B cells from 30 individuals) was imported into R from idat files as a *methylumiSet* using the package *methylumi*[47]. DNAm data underwent stringent quality control following a standard pipeline with the following steps: 1) calculation of median methylated and unmethylated signal intensities excluding samples with median intensity < 1500 (8 samples excluded), 2) using the control probes to calculate a bisulphite conversion statistic using the *bscon* function from the *wateRmelon* package

[48], excluding any sample < 80%, 3) principal component analysis (PCA) of the DNAm data identified that the 6th and 7th PCs separated samples into two groups which correlated with recorded sex to (2 samples that grouped with the incorrect sex were excluded), 4) comparing profiles across the 59 SNP probes on the EPIC array to confirm all matched samples from the same individual were genetically identical, 5) comparing profiles across the 59 SNP probes on the EPIC array across twin pairs to confirm they were genetically identical (one twin pair was found to be dizygotic but was retained in the sample for this study), 6) comparing profiles across the 59 SNP probes on the EPIC array with match SNP chip genotype data for these samples to confirm no sample switches, 7) cellular composition of each sample was estimated using the *estimateCellCounts* function in the *minfi[49]* package excluding supposedly purified cell types that were estimated to consist of < 70% of that cell type (4 samples from the same individual were excluded), 8) using the *pfilter* function from the *wateRmelon[48]* package to exclude samples with >1% of sites with a detection p-value >0.05 (3 samples) in addition to sites with a beadcount <3 in >5% of samples (204 sites) where >1% of samples had a detection p-value >0.05 were removed (11,793 sites). The final dataset included 217 samples (28 buccal, 19 nasal, 29 whole blood, 28 monocytes, 29 granulocytes, 28 CD4+ T cells, 28 CD8+ T cells, and 28 B cells). For every sample, age was predicted from the DNAm data using Horvath Epigenetic clock algorithm[22] implemented with the *agep* function in the *wateRmelon[48]* package and a smoking score was calculated based on DNAm at sites known to be associated with smoking as previously described by Elliot et al[4]. Data was quantile normalised using the *dasen* function from the *wateRmelon* package[48]. Prior to analysis, the 59 SNP probes, sites on the sex chromosomes, sites containing common SNPs[50], and sites with non-specific binding were removed, leaving 784,726 DNAm sites. These DNAm data are publicly available via GEO accession number GSE166844.

## Data analysis

All analyses were performed using DNA methylation measured using beta values (i.e. as a proportion). To identify differentially methylated positions (DMPs)–i.e. DNAm sites characterized by different levels of DNAm across the eight sample types—we fitted a mixed effects model with nested random intercepts for family and individual using the R package *lme4* and *lmerTest* and performed an ANOVA. The ANOVA p-value (determined from an F-test) was used to identify significant DMPs, in addition, t-statistics and p-values comparing each sample type to whole blood (set as the reference category) were used to identify which sample types significant differences were identified within. To identify differentially variable positions (DVPs)–i.e. DNAm sites that differ in their variance of DNAm across all eight sample types— we performed Levene's test. To calculate the level of covariation between sample types, pairwise Pearson's correlation coefficients were calculated between DNAm level in our selected 'reference' tissue (whole blood) and DNAm level in each of the seven other sample types collected from matched individuals; the values were squared and multiplied by 100 to obtain the percentage of variance explained for each site. To calculate the proportion of variance in whole blood explained by the five purified blood cell types, a linear model with whole blood DNAm as the outcome predicted by the DNAm level of each of the five cell types included as covariates was fitted and the $R^2$ value of the full model calculated. To quantify how specific variation in DNAm was to a single blood cell type, we defined "characteristic scores" as follows. Using just the data from the five purified blood cell types, DNAm values at all autosomal probes where adjusted for differences in mean level of DNAm between blood cell types, by taking the residuals from a linear model where DNAm was regressed against cell type. Characteristic scores for each DNAm site and cell type were then calculated by fitting a one-sided Levene's

test comparing the variation of a single cell type against the variation across all samples from the other four cell types, specifically testing for a larger variance in that cell type (i.e. one–tailed test). DNAm sites were determined to be characteristic of single cell type if the P-value from Levene's test was $< 9\text{x}10^{-8}$. Code to reproduce the key analyses are available via GitHub (https://github.com/ejh243/CovariabilityPeripheralTissuesCellTypes).

## Supporting information

**S1 Fig. Heatmap of the first ten DNA methylation principal components across the five purified blood cell types and three peripheral tissue samples (whole blood, buccal epithelial cells and nasal epithelial cells) profiled in this study.** Shown is the mean principal component value for samples grouped by cell- or tissue-type. Each row represents a principal component (PC1 to PC10), with the percentage of variance in DNA methylation by each explained in brackets. Each column represents a single sample type.
(PDF)

**S2 Fig. Density plot of DNA methylation levels across the 784,726 autosomal DNAm sites included in our analysis for each sample type.** Shown is the mean level of DNAm at each site across all individuals.
(PDF)

**S3 Fig. Bar chart showing the proportion of differentially methylated positions (DMPs) compared to whole blood shared between different sample types.** For each sample type the sites identified as differentially methylated relative to whole blood were categorized into those that are uniquely different in that sample type or shared with at least one other sample type. Unique DMPs were defined as those where the t-statistic comparing each sample type to whole blood were significant for only a single sample-type. Bar chart **A)** shows the number and **B)** shows the percentage of unique and shared DMPs compared to whole blood for each sample type.
(PDF)

**S4 Fig. Histogram of the number of sample types in which each DMP is differentially methylated compared to whole blood.** Taking all sites identified as having a significantly different level of DNA methylation compared to whole blood in at least one sample type (n = 611,070, ANOVA $P < 9\text{x}10^{-8}$) we counted the number each of individual sample types characterized by differential DNAm ($P < 0.05$).
(PDF)

**S5 Fig. Heatmap showing the overlap between sample-types for all identified differentially methylated positions.** Taking all sites identified as having a significantly different level of DNA methylation compared to whole blood in at least one sample type (n = 611,070; ANOVA $P < 9\text{x}10^{-8}$) we counted the number each of individual sample types characterized by differential DNAm ($P < 0.05$). Each box in this heatmap represents the percentage of significant DMPs that are shared between two sample types.
(PDF)

**S6 Fig. Histogram showing the most common intersects between sample-types for all differentially methylated positions.** Considering all sites identified as having a significantly different level of DNA methylation in at least sample type compared to whole blood (n = 611,070; ANOVA $P < 9\text{x}10^{-8}$) we considered t-statistics to identify the specific sample types characterized by differential DNA methylation. Shown are the combinations of sample types with the most shared DMPs, with the vertical histogram at the top indicating the number of shared

DMPs and the matrix underneath highlighting specific combinations of sample type. The colored bars in the horizontal histogram in the bottom left indicate the total number of DMPs for each sample type.
(PDF)

**S7 Fig. Density plot of the variation in DNA methylation for each sample-type.** Shown across all autosomal DNAm sites included in our analysis is the distribution of the standard deviation in DNAm at each site. Each sample-type is represented by a different coloured line. Our results show that in general, DNA methylation measured in buccal (purple) or nasal (blue) epithelial samples is more variable across individuals than DNA methylation measured in whole blood and individual constituent blood cell types.
(PDF)

**S8 Fig. Scatterplot comparing the site-specific variance in DNA methylation between different sample-types.** Shown is the standard deviation in DNA methylation for all autosomal DNAm sites included in our analysis for each pairwise combination of sample types.
(PDF)

**S9 Fig. Density plot of the variation in DNAm for each sample-type for differentially variable sites.** Each sample-type is represented by a different colored line. This plot shows that sites with significant variance across sample types are generally characterized by increased variance in buccal (purple) and nasal (blue) epithelial samples compared to whole blood and individual constituent blood cell types.
(PDF)

**S10 Fig. Scatterplot of the site-specific variance in DNA methylation between different sample types across DNAm sites with significantly different levels of variation (n = 194, 247).** Above each plot is the Pearson correlation coefficient.
(PDF)

**S11 Fig. Correlation in the variance of DNA methylation between all sample type combinations for DNAm sites that vary across sample types (n = 196,104).**
(PDF)

**S12 Fig. Inter-individual variation in DNA methylation in whole blood is correlated with variation in isolated blood cell types.** Histograms showing the distribution of correlation coefficients between DNA methylation in whole blood and the five blood cell types. **A)** B-cells, **B)** CD4 T-cells, **C)** CD8 T-cells, **D)** monocytes and **E)** granulocytes. The vertical blue dashed line indicates a correlation coefficient of zero. For all five cell types the distribution of correlation coefficients is skewed to the right.
(PDF)

**S13 Fig. Covariation in DNA methylation between whole blood and individual blood cell-types is higher in the subset of DNAm sites classed as having 'intermediate' levels of DNAm or being highly 'variable'.** Shown are boxplots of variance explained in whole blood for each cell type separately where DNAm sites are split by mean DNA methylation level (x-axis, left panels) and variability (right panels).
(PDF)

**S14 Fig. Inter-individual variation in DNA methylation in whole blood is highly correlated with variation in isolated blood cell types for sites under strong genetic control.** Histograms showing the distribution of correlation coefficients between DNA methylation in whole blood and the five blood cell types the subset of sites where whole blood DNAm is under

strong genetic control (additive genetic effects on DNAm > 0.80, n = 6,617 sites) using esti-mates from Hannon et al[25]. **A)** B-cells, **B)** CD4 T-cells, **C)** CD8 T-cells, **D)** monocytes and **E)** granulocytes. The vertical blue dashed line indicates a correlation coefficient of zero. For all five cell types the distribution of correlation coefficients is dramatically skewed to the right. (PDF)

**S15 Fig. Variation in DNA methylation in whole blood as a predictor of variation in the isolated blood cell types and other peripheral tissues across sites at which DNAm is under strong genetic control.** Shown for the subset of sites where whole blood DNAm is under strong genetic control (additive genetic effects on DNAm > 0.80, n = 6,617 sites) using esti-mates from Hannon et al[25] is the proportion of sites (y-axis) for which variation in whole blood DNA methylation explains different levels of variance in five blood cell types (mono-cytes, granulocytes, CD4$^+$ T cells, CD8$^+$ T cells and B cells) isolated from the same individuals. (PDF)

**S16 Fig. Inter-individual variation in DNA methylation in whole blood is highly correlated with variation in isolated blood cell types for sites associated with an mQTL variant.** Histo-grams showing the distribution of correlation coefficients between DNA methylation in whole blood and the five blood cell types the subset of sites (n = 147,683 sites) where whole blood DNAm is associated with an mQTL variant using data from Hannon et al[26]. **A)** B-cells, **B)** CD4 T-cells, **C)** CD8 T-cells, **D)** monocytes and **E)** granulocytes. The vertical blue dashed line indicates a correlation coefficient of zero. For all five cell types the distribution of correlation coefficients is skewed to the right. (PDF)

**S17 Fig. Variation in DNA methylation in whole blood as a predictor of variation in the isolated blood cell types and other peripheral tissues across sites at which DNAm is associ-ated with an mQTL variant.** Shown for the subset of sites (n = 147,683 sites) where whole blood DNAm is associated with an mQTL variant using data from Hannon et al[26] is the pro-portion of sites (y-axis) for which variation in whole blood DNA methylation explains differ-ent levels of variance in five blood cell types (monocytes, granulocytes, CD4$^+$ T cells, CD8$^+$ T cells and B cells) isolated from the same individuals. (PDF)

**S18 Fig. Inter-individual variation in DNA methylation in whole blood is highly correlated with variation in isolated blood cell types for sites strongly influenced by non-shared envi-ronmental factors.** Histograms showing the distribution of correlation coefficients between DNA methylation in whole blood and the five blood cell types the subset of sites where whole blood DNAm is strongly influenced by non-shared environmental factors (non-shared envi-ronmental effects on DNAm > 0.80, n = 143,501 sites) using estimates from Hannon et al[25]. **A)** B-cells, **B)** CD4 T-cells, **C)** CD8 T-cells, **D)** monocytes and **E)** granulocytes. The vertical blue dashed line indicates a correlation coefficient of zero. (PDF)

**S19 Fig. Variation in DNA methylation in whole blood as a predictor of variation in the isolated blood cell types and other peripheral tissues for sites strongly influenced by non-shared environmental factors.** Shown for the subset of sites where whole blood DNAm is strongly influenced by non-shared environmental factors (non-shared environmental effects on DNAm > 0.80, n = 143,501 sites) using estimates from Hannon et al[25] is the proportion of sites (y-axis) for which variation in whole blood DNA methylation explains different levels of variance in five blood cell types (monocytes, granulocytes, CD4$^+$ T cells, CD8$^+$ T cells and B

cells) isolated from the same individuals.
(PDF)

**S20 Fig. Inter-individual variation in different blood cell types predicts inter-individual variation in whole blood at the same sites.** Scatterplots comparing blood-cell type correlations between cell types. The colour of the point indicates the density of observations at that position ranging from gray (low) to yellow (high).
(PDF)

**S21 Fig. Histogram of variance explained in whole blood by all five cell types combined.**
(PDF)

**S22 Fig.** Boxplots of variance in DNA methylation explained in whole blood by all five cell types combined stratified by A) mean DNA methylation level and B) the variability in DNA methylation at that site.
(PDF)

**S23 Fig. Histogram showing the number of individual blood cell types explaining at least 20% of the variance in DNA methylation in whole blood.**
(PDF)

**S24 Fig. Heatmap showing the ratio of observed to expected number of characteristic sites for each blood cell type across DMPs associated with a range of traits.** Association results were downloaded from the EWAS catalog (http://ewascatalog.org/) and filtered to those identified in whole blood at a significance threshold of P < 1e-7. This heatmap contains all traits characterized by at least five significant associations that were characteristic of a blood cell type. Grey indicates that there was no characteristic sites for that cell type for that trait.
(PDF)

**S1 Table. A summary of demographic variables for individuals included in this study.**
(XLSX)

**S2 Table. Mean differences in DNAm age estimates relative to whole blood generated across the different sample-types profiled from each individual.**
(XLSX)

**S3 Table. Mean differences in DNAm-derived smoking estimates relative to whole blood generated across the different sample-types profiled from each individual.**
(XLSX)

**S4 Table. Number of DNAm sites in each blood cell-type at which that cell-type is the major driver of variation detected in whole blood.**
(XLSX)

**S5 Table. Differentially methylated sites associated with BMI demonstrating characteristic variation of a single blood cell type.**
(XLSX)

**S6 Table. Differentially methylated sites associated with tobacco smokingI demonstrating characteristic variation of a single blood cell type.**
(XLSX)

**S7 Table. Differentially methylated sites associated with traits in the online EWAS catalogue demonstrating characteristic variation of a single blood cell type.**
(XLSX)

**S8 Table. Summary of the mean number of individual blood cell-types collected by FACS from whole blood.**
(XLSX)

## Author Contributions

**Conceptualization:** Eilis Hannon, Jonathan Mill.

**Data curation:** Eilis Hannon, Emma Walker, Joe Burrage, Agnieszka Kepa, Janis Best-Lane.

**Formal analysis:** Eilis Hannon, Georgina Mansell, Emma Walker, Marta F. Nabais.

**Funding acquisition:** Terrie E. Moffitt, Avshalom Caspi, Louise Arseneault, Jonathan Mill.

**Investigation:** Joe Burrage, Agnieszka Kepa, Suzanne Heck.

**Methodology:** Eilis Hannon.

**Project administration:** Janis Best-Lane, Louise Arseneault, Jonathan Mill.

**Resources:** Anna Rose, Suzanne Heck.

**Software:** Eilis Hannon, Georgina Mansell, Marta F. Nabais.

**Supervision:** Jonathan Mill.

**Validation:** Emma Walker, Marta F. Nabais.

**Visualization:** Eilis Hannon, Georgina Mansell, Emma Walker.

**Writing – original draft:** Eilis Hannon, Jonathan Mill.

**Writing – review & editing:** Terrie E. Moffitt, Avshalom Caspi, Louise Arseneault.

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
