## [Decision Letter · Decision Letter 0]

10 Aug 2020

Dear Dr Hannon,

Thank you very much for submitting your Research Article entitled 'Assessing the co-variability of DNA methylation across peripheral cells and tissues: implications for the interpretation of findings in epigenetic epidemiology' to PLOS Genetics. Your manuscript was fully evaluated at the editorial level and by independent peer reviewers. The reviewers appreciated the attention to an important problem, but raised some substantial concerns about the current manuscript. Based on the reviews, we will not be able to accept this version of the manuscript, but we would be willing to review again a much-revised version. We cannot, of course, promise publication at that time.

If you decide to revise the manuscript for further consideration at PLOS Genetics, please aim to resubmit within the next 60 days, unless it will take extra time to address the concerns of the reviewers, in which case we would appreciate an expected resubmission date by email to plosgenetics@plos.org.

[LINK]

We are sorry that we cannot be more positive about your manuscript at this stage. Please do not hesitate to contact us if you have any concerns or questions.

Yours sincerely,

John M. Greally, D.Med., Ph.D.

Section Editor: Epigenetics

PLOS Genetics

John Greally

Section Editor: Epigenetics

PLOS Genetics

The reviewers all find that the study would be of interest to the readers of PLoS Genetics. However, they raise concerns regarding the statistical analyses, how well the results are placed in the context of the field, and the general accessibility of the conclusions from the work. Each of these are key areas that need to be addressed.

Reviewer's Responses to Questions

**Comments to the Authors:**

Reviewer #1: Hannon and colleagues generated genome-wide DNA methylation data in main blood cell types, whole blood, nasal epithelial cells, and buccal epithelial cells to address various important questions related to epigenome-wide association studies (EWAS). Such studies are often performed in whole blood and any association observes may therefore stem from a DNA methylation change involving a single blood cell type, involving multiple/all cell types, or not a change in DNA methylation itself but a shift in cell type proportions. The authors present various analysis on differences in mean and variance and on covariation to gauge the impact of studying whole blood in EWAS. Although the analyses as presented are interesting, the power of the data set and the relevance of the questions raised for the EWAS-field have the potential to lead to more defined and specific insights as to the interpretation of EWAS. The authors are to be commended for already making available these important data though GEO both as idat files and processed data.

- I missed whether beta-values or M-values were used. Since variances are compared between cell types and the means between cell types substantially differ, heteroscedasticity is a real issue here and M-values should be used. If they were not used, the current results likely over-estimate differences in variance between cell types.

- The title reads “implications for the interpretation of findings in epigenetic epidemiology”. Indeed, I think this is the main interest of the manuscript. Can the authors sum up the specific implications for the interpretation in the discussion?

- To find Cell and tissue specific DNA methylation an ANOVA test was used. It remained unclear how the test accounted for the fact that donors provided multiple samples and that the donors were related (twins). If a random effects model was used, this would have been immediately clear. Also, instead of a post-hoc test, the authors perform T-tests after the ANOVA.

- It would be helpful for the reader to include a table that shows the percentages (mean and SD) of the cell types based on flow sorting. Also, inclusion of some other epidemiological data will be helpful (sex, bmi, smoking etc).

- How does the percentage of differentially methylated CpGs compare to previous studies? I was surprised to see such a large percentage of CpGs to be tissue/cell specific (~75% of CpGs tested). As noted by the authors the array is biased towards CpG islands which tend to be unmethylated independent of cell type. I realize that testing or correcting for statistical inflation is not possible here. The inclusion of sensitivity tests to gauge the validity of the finding would be in order.

- The application of the pan-tissue clock is valuable (which apparently is not as pan-tissue as sometimes presumed). The discussion may mention that other clocks developed for specific tissues and with better performance (for example due to the use of a larger test and training set) may lead to more accurate predictions.

- The authors compare variances of CpGs across tissues. I find it difficult to see what specific recommendation for EWAS can result from this analysis. This section reads as a very statistical and more concrete examples can help the reader to see the relevance. Note, that the use of M-values is critical here. Even if variances are different, the correlation can be high? Also, much of the variance, particularly when beta values are lower, actually is technical noise in the array. If the noise is CpG-specific, this may suggest biological similarities between tissues that are actually related to the platform. One possibility to increase the relevance of this section could be to focus on hits from major whole blood EWAS for which reliable meta-analysis were published (including but not limited to smoking, age, lipids, BMI, CRP…). However, a major omission here is the inclusion of genetic data. Methylation QTLs will be a main driver of similar variance between tissues. Evaluating variances after regressing out main cis-QTL effects (taking SNPs previously reported for whole blood) will lead to much more informative data. Generating SNP data for 16 individuals is quite cheap.

- Scrutinizing previous whole-blood EWAS hits (including but not limited to smoking, age, lipids, BMI, CRP; the first rheumatoid arthritis EWAS, which was extremely confounded by cell type, may be a nice one too) would significantly strengthen the message of the manuscript. The data should allow the authors to say whether or not it is likely whether or not findings were cell type-specific. In the discussion that fact that samples were European-descent and 19 years of age is presented as caveat. While this may be true for an epidemiological study, here more fundamental biology is addressed (cell-specific DNA methylation) for which such factors are less important. The authors already look in general terms at CpGs reported for smoking and BMI but the data has much more potential.

- Indeed, the cumulative variance of specific cell types is the factor explaining whole blood variance. Including the measured percentage per cell type per individual will provide a more exact view of what happens and improve interpretation.

- I was confused by the “characteristic scores”. I see that the analysis results in a set of CpGs that to some extent have a specific DNA methylation pattern for a certain cell type. But due to the approach taken I do not see how this can be used by other studies. I think the authors have great data to come up with a superior deconvolution algorithm, but the characteristic scores cannot be used for that. I guess my problem is that the approach does not yield a comprehensive set of CpGs that are cell type specific and also it does not yield a very robust set of CpGs that can be used in external studies. Can the authors use more established methods aimed at prediction?

- The authors present the overlap between individual CpGs reported on in EWAS of smoking and BMI and overlapping with the characteristic score and then conclude that a specific cell type was driving the EWAS hit. I think this kind of analysis potentially is the most interesting part of the manuscript and indeed figure 6 is very compelling. But if a shift in for example B-cells drove the detection of a specific EWAS finding, then other CpGs should also be detected that stem from B cells. In fact, the authors be have the power to distinguish between EWAS findings that are driven by a shift in cell type proportion and a cell-specific DNA methylation change. The first will lead to a change in DNA methylation for many CpGs with a B-cell-specific DNA methylation level, while the latter will be restricted to only one or a few CpGs. More generally: comparing per CpG as done now and only looking at the CpGs present in the ‘characteristic score’ may not be most informative. Instead, taking whole blood EWAS hits for an outcome as a starting point and study the DNA methylation level of these CpGs are across cell types may give valuable information on the interpretation of EWAS. For such analysis, using the most cell-type specific CpGs as a backgound/reference may be of interest.

Reviewer #2: In this manuscript, the authors have used the EPIC array to analyse methylation profiles in a variety of human tissues from 30 different individuals. I can see some aspects of this work as being useful to groups conducting EWASs, and I think the analyses presented are technically sound. My main concerns lie with the authors’ not really placing their work in the context of what has been done before, which both means the novelty is overstated and discussion of discrepancies or improvement over previous data is quite insufficient. I will present just a few examples:

1. They state this is the largest study of purified blood cell types to date. The Beck lab and colleagues (Paul et al., 2016, Nat Comm), from the BLUEPRINT project, generated data from 52 MZ twin pairs discordant for T1D (so including both T1D and healthy controls) in three immune effector cell types: CD4+ T cells, CD19+ B cells and CD14+CD16− monocytes. They supplemented this with analyses of three additional, genome-wide DNA methylation data sets in CD14+ monocytes and CD4+ T cells from 12 T1D-discordant MZ twin pairs; CD14+ and CD4+ cells from 201 and 139 unrelated, healthy individuals; and cord blood from 98 newborns of whom 50 had progressed to overt T1D during childhood.

Or also Ewing et al., 2019, EBioMedicine. They measured DNA methylation in CD4+ T cells (n = 31), CD8+ T cells (n = 28), CD14+monocytes (n = 35) and CD19+ B cells (n = 27) from relapsing-remitting (RRMS), secondary progressive (SPMS) patients and healthy controls (HC) using Infinium HumanMethylation450 arrays. Monocyte (n = 25) and whole blood (n = 275) cohorts were used for validations.

2. They make several conclusions about how ageing epigenetic clock varies in different tissues. This is hardly new. I suggest referring to several reviews on this topic: Adam et al., 2019, Mol Cell; Bell et al., 2019 Genome Biology; Horvath and Raj, 2018, Nature Reviews Genetics, and the primary papers quoted in these reviews. The fact that the clock doesn’t work equally well in different tissues is hardly new knowledge. Also, the fact that the authors samples were 19 years old only represents a major limitation as it is possible that better correlations are observed in older individuals.

3. There is significant discussion of the peculiarities of blood DNA meth profiles vs other tissues. I would encourage the authors to look at Varley et al., 2013, Genome Research; Lowe et al., 2015 Epigenetics).

Reviewer #3: The manuscript by Hannon and colleagues characterises difference in DNA methylation signal from multiple cell types from the same donors, in order to investigate the extent to which tissue-specificy impacts DNA methylation measurements, and be able to recommend approaches to disambiguate possible confounders in EWAS.

I find the work of broad interest, and I appreciate the note of caution it introduces to the interpretation of EWAS studies, especially those that compare DNA methylation from different tissue types. The paper itself is well-written and its conclusions carefully considered; I appreciated the nuance to the discussion and the straightforward introduction. I do, however, also have some serious methodological concerns that I detail below, and which I believe have the potential to impact some of the conclusions of the paper.

Major comments:

1. Samples and data processing: In the methods (line 473), the authors state that their samples do not represent 30 unrelated individuals, as one might have assumed to that point, but rather 15 pairs of twins - 14 monozygotic, and 1 dizygotic (presumably a surprise). The nested structure of the data does not seem to be controlled for at any point, which is rather worrisome in a manuscript that is built around estimates of variance; these 30 samples do not represent independent observations, as the methods the authors describe assume. Correcting for this structure in all analyses is *essential* to drawing robust conclusions, especially from line 261 onwards, where the work begins to focus on inter-individual differences. Alternatively, consider simply dropping one sample from each twin pair.

I also noted that authors processed samples in multiple batches (line 509, "Samples from the same individual were processed together across all experimental stages to negate any methodological batch effects"); it would be nice to know how many samples per batch and the design of batches/arrays, and to include it as a factor in all testing models, as their design may have accidentally exacerbated inter-individual differences. Age and sex should also be included as covariates any time linear models are being fit; it's unclear this is taking place.

Additionally, the authors remove 59 probes that span known SNPs and are used for genotyping; I encourage them to remove all probes that overlap known SNPs segregating within European pops (1000G or Hapmap, does not matter) at appreciable frequencies, as these too can impact methylation measurements. See, eg, https://www.ncbi.nlm.nih.gov/pmc/articles/PMC6923858/ (no need to use this package).

2. Differential methylation testing, lines 196-266: The authors use pairwise ANOVA to identify DMPs between whole blood and all other tissues under consideration. This is not a robust approach, and there are many better suited methods for DMP identification, including missmethyl, methylKit and even limma (a wrapper for the limma lmFit function is provided in minfi), to name a few, that will leverage the information from the entire dataset across all pairwise contrasts.

Most of these packages have easily accessible tutorials so this should hopefully not be too onerous; regardless of the one the authors choose to use, they should, as above, redo the computation of DE accounting for the lack of independence in their dataset, controlling for both the relatedness between sample pairs as well as the donor effect across the multiple tissues being tested. As above, much more complex experimental design than suggested is needed for the robust identification of DMPs.

Finally, the p-value threshold of 9x10^-8 for significance is slightly more permissive than Bonferroni correction for the number of tests being performed (0.05/784726); I encourage the authors to consider FDR approaches implemented in limma and other packages instead, which are more typical in DM testing.

3. Line 221, "Overall, our results suggest that cell-specific levels of DNAm occur in a hierarchical manner, where at a subset of genomic loci profiles are shared between cells from the same linage." I find this to be an intriguing observation, and would like to see more; I think the overlap in DMPs discussed here and captured, eg, in Supp Fig 3 and 4, would be much better appreciated in a different format - potentially an upsetR plot or similar?

4. Line 236 onwards: It is not clear when reading this section that while the mean difference in age when talking about whole blood (line 236) is given as a difference to the donors' actual age, the rest of them are reported as differences with the whole-blood estimate. This makes the section confusing to read, and makes many of the predictors appear to perform far more poorly than expected. Please report all numbers as differences against the same value, or clarify this further.

5. At the beginning of the results (line 154) the authors state "Following pre-processing, normalization and stringent quality control (see Materials and Methods) our final dataset included measures of DNAm at 784,726 autosomal sites [...]"; in the methods (line 541) they state that "Prior to analysis, the 59 SNP probes, sites containing common SNPs, and sites with non-specific binding were removed, leaving 802,216 DNAm sites."

Which one is the correct number, and why the discrepancy? Is the difference simply autosomal vs genome-wide? If so, when were the X chr probes removed - before or after normalisation?

Minor comments:

Line 169 - principal component (PC) analysis was used to determine the optimal axes of variation - what exactly do the authors mean by 'optimal'? This is a strange term in this context.

Genetic impacts on DNA methylation level: I found this section somewhat dismissive of the impact of genetic effects on inter-individual differences in DNA methylation; the authors take only those probes that exhibit a near perfect correlation across tissues (line 308) as driven by genetic differences, but I find this very conservative - I would argue that there is scope in this section for either more nuance in the analyses, but I leave it to the authors to decide.

All figures - the chosen colour scheme (default R colours) is hard to visualise on some screens/printouts (yellow especially); I suggest the authors consider choosing slightly more display friendly colours (eg: https://personal.sron.nl/~pault/), although this is very much an optional suggestion

Figure 1a - whole blood is not labelled in the legend.

Figure 1b - I found the 3d plots very hard to interpret. Since what is being plotted are actually the three possible pairwise combinations of PCs 1, 2, and 6 it would be much better to replot these as 2d, showing one pairwise-comparison at a time.

Supp fig 7 - are the vertical white lines in the second row of plots compression artefacts (in which case I might be the only one seeing them!) or actual signal? If the latter, what is driving these trends?

**Have all data underlying the figures and results presented in the manuscript been provided?**

Reviewer #1: Yes

Reviewer #2: Yes

Reviewer #3: Yes

PLOS authors have the option to publish the peer review history of their article (what does this mean?). If published, this will include your full peer review and any attached files.

Reviewer #1: No

Reviewer #2: No

Reviewer #3: No

---

## [Decision Letter · Decision Letter 1]

11 Dec 2020

Dear Dr Mill,

Thank you very much for submitting your Research Article entitled 'Assessing the co-variability of DNA methylation across peripheral cells and tissues: implications for the interpretation of findings in epigenetic epidemiology' to PLOS Genetics.

The manuscript was fully evaluated at the editorial level and by independent peer reviewers. The reviewers appreciated the revisions made according to their first reviews, and your attention to an important topic, but as you will see, two of the three reviewers identified some remaining concerns that we ask you address in a revised manuscript.

We therefore ask you to modify the manuscript according to the review recommendations. Your revisions should address the specific points made by each reviewer.

[LINK]

Yours sincerely,

Marnie E. Blewitt

Associate Editor

PLOS Genetics

John Greally

Section Editor: Epigenetics

PLOS Genetics

Reviewer's Responses to Questions

**Comments to the Authors:**

Reviewer #1: I much appreciate the significant effort the authors put in the revision. They added various interesting analyses and included valuable remarks to guide the interpretation.

My main concern continues to be the biological relevance of the variance analysis and the role of heteroskedasticity. The point is that CpGs with a mean methylation (beta value) closer to 0.5 will be more variable not due to biological influences but a mathematical effect. For me the clearest example is that most meQTLs will, apart from showing statistically different methylation between genotypes, also show a difference in variance. The latter is not due to additional environmental influences, but just the mathematical phenomenon. I strongly disagree with the authors’ assumption that a change in variance equals biology and the interpretation that it reflects differential effects of environment is speculative if not incorrect (“However, this reflects biology” / “We identify numerous differences in the variability of DNAm between tissue and cell types, indicating that differences in DNAm (induced by environmental exposures, for example) have variable effects across cell types and tissues.”). In fact, Levene’s test is sensitive to the mathematical source of heteroskedasticity. One way out would be to instead apply double generalized linear models (DGLM) which is specifically developed to address the issue.

The author state that: “The goal of our analyses was not to identify the source of the variation we observe, but to characterise and describe how differences are reflected across sample types.” However, making the distinction between genetic and non-genetic sources seems crucial, in particular because the authors several times speculate about the role of environmental influences as an explanation for their findings. For example, genetic effects are likely to increase while environmental factors are likely to decrease correlation between tissues since the former is a pan-tissue phenomenon and an environmental; factor commonly affects a subset of tissues. If the authors are unable to measure a SNP array in the 30 individuals included in their analysis to directly correct for cis-methylation QTL effects, another indirect assessment is required.

In the Discussion the authors state ‘therefore, isolating populations of cells is required to improve our understanding of the mechanisms underlying epigenetic dysregulation’. I think a more balanced conclusion is warranted on the basis of the data since the large majority of EWAS findings was not related to a cell-type specific effect according the analyses presented in the manuscript.

May the fact that methylation in the major blood cell types explain only a proportion of whole blood methylation reflect the limitation mentioned in the discussion that many more blood cell (sub)types exist with varying abundance between individuals and DNA methylation levels? I guess that since whole blood methylation variation is not captured by main blood cell types, the main blood cell methylation variation does not fully capture variation in sub cell types.

Reviewer #2: The manuscript is much improved and I have no major issues.

Reviewer #3: I thank the authors for their thorough response to my comments. I enjoyed rereading the manuscript, and the nuanced interpretation of the data presented within. As far as the figures go, the new colour scheme is much improved and easier to read, and the new 2D PCA plots look great and are also easier to interpret. I do have a couple of comments still:

Batch effects and processing:

I thank the authors for their clarification, but I am concerned by the strategy they reveal, which sets them up to confound technical and biological variation. From their answer to my original comment, samples from each twin pair were processed on different arrays, with samples from the same tissue from a single twin pair run on the same array. This means that if, eg, twin pair 5 is consistently different from all other twin pairs it is impossible to disambiguate whether the effect is driven by biology or technology. (For instance the couple of clustering failures and outliers in Fig1A in whole bloood and nasal epithelium come in pairs - where they from the same ind/array?)

I realise nothing can be done about it at this point, but I am curious to know if the authors included processing batches/array batches as covariates in their lme models. I also urge them to instead randomise their samples next time around, and to ensure their technical covariates are orthogonal to their variables of interest, for even more robust results.

I realise that was more of a comment than a question.

Test statistics:

The moderation of the multiple testing still seems insufficient to me. The authors perform ~700k lm/ANOVA (one per probe), then multiple t-tests (whole blood vs the seven other tissues) for those probes that ANOVA deems significant (~600k) to identify the relevant pairwise comparisons driving it, so 700k + (7*600k), which is a lot of tests. But the significance threshold for the t-test is set to 0.05 (inferred from one of the supp figure legends, since it is not reported in the methods) instead of 0.05/7 or similar. I note that in their response to me the authors claim that Tukey's HSD or similar would test too many comparisons (I don't necessarily disagree, it's a lot of pairwise comparisons!), but the current approach does not seem to me like a robust posthoc approach, and I would like to see more details on why the authors deem it appropriate to not even moderate the t-test p vals.

line 175: The authors should state here that their samples come from twin pairs. This colours the reading of all subsequent results. This should also be made clearer in line 649.

line 582: alluded, not eluded, I hope!

line 774: publically should be publicly

line 774: I looked up the GEO accession and find it only leads to the data from the purified blood cells, but not to the whole blood, nasal or buccal cells?

Figure 1:

I couldn't help but notice that the clustering is ever-so-slightly different between the version of the figure in the original submission and in this current version. Anything interesting happening there?

Figure 2:

Totally optional but a dashed line at 19 indicating the true sample age might be a nice touch.

**Have all data underlying the figures and results presented in the manuscript been provided?**

Reviewer #1: Yes

Reviewer #2: Yes

Reviewer #3: **No: **Missing some data from the supplied GEO accession number, unless I'm missing something obvousl

PLOS authors have the option to publish the peer review history of their article (what does this mean?). If published, this will include your full peer review and any attached files.

Reviewer #1: No

Reviewer #2: No

Reviewer #3: No

---

## [Editor Report · Decision Letter 2]

23 Feb 2021

Dear Dr Mill,

We are pleased to inform you that your manuscript entitled "Assessing the co-variability of DNA methylation across peripheral cells and tissues: implications for the interpretation of findings in epigenetic epidemiology" has been editorially accepted for publication in PLOS Genetics. Congratulations!

Yours sincerely,

Marnie E. Blewitt

Associate Editor

PLOS Genetics

John Greally

Section Editor: Epigenetics

PLOS Genetics

Comments from the reviewers (if applicable):

**Data Deposition**

http://datadryad.org/submit?journalID=pgenetics&manu=PGENETICS-D-20-00987R2

**Press Queries**

---

## [Editor Report · Acceptance letter]

14 Mar 2021

PGENETICS-D-20-00987R2 

Assessing the co-variability of DNA methylation across peripheral cells and tissues: implications for the interpretation of findings in epigenetic epidemiology 

Dear Dr Mill, 

We are pleased to inform you that your manuscript entitled "Assessing the co-variability of DNA methylation across peripheral cells and tissues: implications for the interpretation of findings in epigenetic epidemiology" has been formally accepted for publication in PLOS Genetics! Your manuscript is now with our production department and you will be notified of the publication date in due course.

With kind regards,

Alice Ellingham

PLOS Genetics

On behalf of:
